# Simulation and Experiment Analysis of Temperature Field of Magnetic Suspension Support Based on FBG

**DOI:** 10.3390/s22124350

**Published:** 2022-06-08

**Authors:** Huachun Wu, Cong Huang, Ruifang Cui, Jian Zhou

**Affiliations:** 1School of Mechanical and Electronic Engineering, Wuhan University of Technology, Wuhan 430070, China; whcwhut@whut.edu.cn (H.W.); 280905@whut.edu.cn (C.H.); 18372039940@163.com (R.C.); 2Hubei Provincial Engineering Technology Research Center for Magnetic Suspension, Wuhan 430070, China

**Keywords:** magnetic suspension support, temperature field, FBG temperature sensor, equivalent model

## Abstract

Temperature rise is an important factor limiting the development of magnetic suspension support technology. Traditional temperature sensors such as thermocouples are complicated and vulnerable to electromagnetic interference due to their point contact temperature measurement methods. In this paper, the equivalent model of magnetic suspension support is established, and the temperature field is simulated and analyzed by magnetic thermal coupling calculation in ANSYS software. Then, a quasi-distributed temperature measurement system is designed, and the FBG temperature sensor is introduced to measure the temperature of the magnetic suspension support system by “one-line and multi-point”. By comparing the analysis experiments and simulations, the equivalent accuracy of the simulation model and the FBG temperature sensor can accurately measure the temperature of the magnetic suspension support.

## 1. Introduction

Magnetic suspension support technology uses electromagnetic force to realize the contactless suspension of the suspended body. Compared with the traditional mechanical support technology and air suspension support technology, it has many advantages and is widely used in magnetic suspension trains, magnetic suspension bearing and magnetic suspension vibration isolation devices [1,2]. However, with the increase in the operation energy consumption of the magnetic suspension support system (MSSS), the temperature of the system rises. The temperature rise mainly has the following effects on the MSSS: (1) It affects the reliability of the system. When the system temperature exceeds its allowable temperature, it leads to the aging of insulating materials and reduce its service life; (2) When the system temperature is too high, the temperature drift of the sensor affects the accuracy of displacement detection, which further affects the dynamic performance of the whole system. It can be seen that the temperature rise is an important factor limiting the development of suspension support technology. At present, there are many methods of temperature measurement. Thermocouple and thermal resistance are common “point” contact temperature measurement technology. Through this method, the temperature values of multiple positions of the measured object can be obtained. When using this temperature measurement technology to measure the temperature of multiple positions, the monitoring system will have too many wires, which is complicated, and the temperature measurement technology is vulnerable to electromagnetic interference and poor stability. The point thermometer also belongs to the contact temperature measurement technology. When measuring the temperature, it must contact the surface of the measured object. Its measurement accuracy is affected by the roughness of the measured surface and the perpendicularity between the probe and the measured surface, and the accuracy is low; an infrared thermal imager is a typical non-contact temperature measurement technology, which has the advantages of non-contact, fast and high-temperature resolution. However, an infrared thermal imager can only measure the surface temperature field that can be photographed; compared with these temperature measurement technologies, fiber Bragg grating (FBG) temperature sensor has the advantages of small volume, good stability, high measurement accuracy, electromagnetic interference immunity and “one-line and multi-point” distributed online monitoring.

At present, FBG sensing technology is widely used in industry, medicine and other fields. Rao [3] gave a systematic and detailed introduction to FBG sensing technology. Rao, David, Webb et al. [4] applied FBG sensor technology to the medical field and measured the temperature of the human body through the FBG temperature sensor. The experiment shows that the measurement accuracy of the FBG sensor can reach ±0.2 °C in the range of 30–60 °C [5]. Scholars applied FBG sensor technology to the baby delivery room, designed an external real-time monitoring system, used multiple FBG sensors for measurement, optimized the way of measuring and processing data and reduced the measurement error caused by multiple sensors [6]. Scholars presented the detection of flaws in the outer bearing’s raceway from the measurement of motor dynamic strain signals collected from sensors based on fiber Bragg grating (FBG). Aiming to carry out real-time online monitoring of the thermal characteristics and their effect on the machine tool spindle bearing stiffness, a fiber Bragg grating (FBG) sensors network was proposed [7]. The online measurement of oil-film pressure inside a journal bearing based on an optical fiber Bragg grating (FBG) is proposed and described for the first time [8]. The FBG temperature sensors had good accuracy, stability and consistency when measuring the temperature distribution of the bearing bush [9]. Liu et al. [10] proposed a multi-point quasi-distributed sensing method based on embedded fiber Bragg grating (FBG) sensors to measure the temperature field of the outer ring of the bearing under three different working conditions: idling, uploading axial and radial cutting force. Wang et al. [11] modified temperature-compensation function with the influence of interfacial action considered is proposed to enhance the measurement accuracy of FBG-based sensors. From the research status of FBG sensor technology at home and abroad, FBG sensor has been applied to the temperature field measurement of machine tool spindle and bearing and has high measurement accuracy.

At present, many scholars at home and abroad have conducted a lot of research on the temperature rise of magnetic bearing. Mizuno, T. [12] studied the loss of magnetic bearing under the two configurations of magnetic pole NSNS and NNSS. Through experimental verification, it was concluded that the iron loss of magnetic pole of magnetic bearing under NNSS configuration is slightly smaller, and its loss value is also affected by ampere turns, speed, control current and air gap. David C. Meeker et al. [13], taking the homopolar laminated radial magnetic bearing as the research object, proposed the analytical solution of rotor rotation loss and verified the correctness of the analytical solution by comparing the experimental data. Jiang et al. [14] proposed a novel analytical calculation method based on distributed magnetic circuit method (DMCM) to solve the problem of coupled magnetic flux and saturation of active magnetic bearing (AMB). Cao, Y. et al. [15] analyzed the loss and temperature of the hybrid magnetic bearings (HMB), and the results showed that the loss of HMB is mainly distributed in the rotor part, and the temperature of the rotor part is obviously higher than that of the stator part. Li et al. [16] established the thermal model of the motorized spindle and simulated the steady-state temperature field of the motorized spindle by using ANSYS. Zhai et al. [17] established the prediction models of iron loss and copper loss for magnetic bearings and BLDCM and determined the loss coefficients of iron core by measured loss curves at different frequencies. Dong et al. [18] proposed a multi-physical field simulation based on a magneto-thermal-fluid coupled iterative solution. Kondaiah, V.V. et al. [19] proposed an experimental approach to study the loss factor and efficiency of an active magnetic thrust bearing. Zan et al. [20] combined thermal network analysis with the temperature field distribution to obtain the key factors affecting the heat dissipation of the flywheel. Kepsu, D. et al. [21] analyzed the influence of segmentation of the magnets on losses. Zhong Ma et al. [22] used chitosan as a multifunctional layer to create a dual-modal flexible sensor. The frequency dependency of the capacitive output enabled modulations of the responses and a simple yet effective approach for decoupling the pressure and temperature responses with an average accuracy of ~92.3%. Basov, M. [23] developed a small silicon chip of Schottky diode (0.8 × 0.8 × 0.4 mm^3^) with the planar arrangement of electrodes (chip PSD) as a temperature sensor and proved the application of PSD chip for a wider temperature range from −65 to +115 °C. Nuo Zhang et al. [24] demonstrated a high-performance temperature sensor based on a 4H-SiC pn diode at a forward current density of 0.44 mA/cm^2^; the device achieves a sensitivity of 3.5 mV/°C.

From the current research status, it can be seen that the FBG temperature sensor has not been used to monitor its temperature in the research on the temperature field of magnetic suspension support. Compared with other temperature measuring instruments, the FBG temperature sensor has higher measurement accuracy, strong anti-interference and smaller volume. It is just suitable for the occasion of compact structure and high operation accuracy of MSSS. Therefore, an FBG temperature sensor is introduced into the research of the temperature field of magnetic suspension bearing in this paper.

## 2. FBG-Based Temperature Sensors

### 2.1. FBG Sensing Principle for Temperature

As a new type of temperature sensor, the FBG temperature sensor has the characteristics of small and soft optical fiber, electromagnetic interference immunity, strong environmental adaptability, corrosion resistance, long-distance transmission of sensing optical signal, and multiple sensing gratings can be prepared on one optical fiber to realize “one-line and multi-point” signal sensing, breaking through the limitations of few measuring points and single output function of traditional sensors. It has technical advantages and prospects for wide application in the field of multi-parameter distribution dynamic detection.

The structure and principle of FBG are shown in Figure 1. The principle is that when the broadband light source passes through the grating as the incident light, the light wave with the same phase as the grating is reflected, and the other light waves are transmitted from the FBG. The measured physical quantity can be obtained through the light wave reflection theory [25].

According to the principle of FBG optical detection, the reflection wavelength moves to the right or left depending on the measured physical size. The most important measurement of the FBG detection system is wavelength measurement. Then, the wavelength offset is converted into amplitude, phase, frequency and other amplitude, which can be measured more easily. Amplitude measurement is the most common and direct measurement method in optical fiber sensors. The wavelength shift is converted into amplitude change, which makes the detection system simpler and cheaper.

When a broadband light is an incident on the fiber grating, the grating is used as a narrow-band filter. The periodic structure of the refractive index makes the reflection of a specific wavelength, and the central wavelength of the reflected light signal depends on the Bragg wavelength. FBG reflects light at the $\lambda_{B}$ Bragg wavelength:(1)$$\lambda_{B}=2n_{eff}\Lambda ,$$

Here, Λ is the grid period, and the $n_{eff}$ is an effective refractive index. As temperature change with Λ and $n_{eff}$, the wavelength of the reflected light will also change. Therefore, the temperature measurement can be realized by wavelength offset.

When the temperature changes, the wavelength of the emitted light changes [26,27]:(2)$$\frac{\Delta \lambda_{B}}{\Delta T}=2n_{eff}\frac{\partial \Lambda }{\partial T}+2\Lambda \frac{\partial n_{eff}}{\partial T},$$

Combining Equations (1) and (2), we obtain:(3)$$\frac{\Delta \lambda_{B}}{\Delta T}=\frac{1}{\Lambda }\frac{\partial \Lambda }{\partial T}\lambda_{B}+\frac{1}{n_{eff}}\frac{\partial n_{eff}}{\partial T}\lambda_{B},$$

When Equation (3) is rearranged:(4)$$\frac{\Delta \lambda_{B}}{\lambda_{B}}=\left(\alpha +\eta \right)T,$$

Here, the change in $\Delta \lambda_{B}$ is Bragg wavelength, and α and η are thermal expansion and thermo-optic coefficients, respectively. In the normal temperature area, $\eta =6.67\times 10^{-6}\;$°C^−1^. Compared with the coefficient of thermal expansion, the contribution of the thermo-optic effect to the wavelength change is 95%. For fiber gratings with a central wavelength of 1300 nm, a temperature change of 1 °C leads to a wavelength of 9.13 pm.

When multiple FBGs are used in combination, the above equation also applies to a single FBG. However, the bandwidth of FBG needs to be reasonably selected to avoid interference with each other.

### 2.2. Sticking and Calibration of FBG Temperature Sensor

The application of FBG for thermal sensors faces the problem of the cross-impact of temperature and stress. Both can make the wavelength of FBG drift, so in one judge, only considering the wavelength drift, the reason is not clear; it is very difficult to attribute the wavelength change to thermal change or strain change. Therefore, in order to implement the practical use of the FBG thermal sensor, we must choose appropriate packaging technology or eliminate the effect of non-temperature.

Packaging structure [28] pastes the FBG and its both ends in the trough, the pasted points approach both ends, and between the points in one extreme little bend, the pipe is filled with non-solidified thermal paste. When exterior load is imposed on both ends, each pasted point of both ends can bear part of the force; at the same time, the bending part performs a buffer role against the tensile deformation caused by an outside force. In addition, the measured object electromagnet has small stress deformation during operation.

In order to ensure that the measured data of the FBG temperature sensor are more accurate and reliable, the temperature of the FBG sensor must be calibrated in the electrothermal incubator before the experiment, and the relationship between FBG central wavelength and temperature can be obtained through the experiment.

In the experiment, when the data acquisition time is 5 s at a certain temperature, 5000 data can be obtained. The average value of the collected 5000 data is taken as the central wavelength of FBG at that temperature. Each sensor is calibrated twice. The experimental data are processed by the least square method, and the calibration data of each sensor are linearly fitted to obtain the fitted result diagram of each sensor, as shown in Figure 2. Finally, the sensitivity range of the FBG temperature sensor used in the experiment is 9.5 pm/°C~10.5 pm/°C, and the linear correlation is high.

## 3. Basic Structure and Heat Source Analysis of Magnetic Suspension Bearing 

### 3.1. Basic Structure

At present, the MSSS can be divided into electromagnetic suspension (EMS) and electrodynamic suspension (EDS) according to the suspension principle. The levitation principle of Maglev bearing studied in this paper is EMS. Through the real-time measurement of the distance between the train and the track and by adjusting the current in the electromagnet, the controllable electromagnetic force is generated to make the train levitate stably. Figure 3 is the structure diagram of the simple experimental device of magnetic suspension support, which mainly includes E-type suspension electromagnet, electromagnet support, track support, electromagnet shell, suspension support, suspension track, limit screw and eddy current displacement sensor.

### 3.2. Heat Source Analysis

The heating component of magnetic suspension bearing is mainly an electromagnet that plays the role of suspension, and its loss mainly includes copper loss and iron loss.

#### 3.2.1. Copper Loss

Copper loss refers to the Joule heat generated when the coil winding is energized, and its calculation formula is as follows:(5)$$P_{cu}=I^{2}R=I^{2}\cdot n\frac{\rho_{cu}l}{A_{cu}},$$

Here, I is the current through the coil winding of electromagnetic iron; R is copper conductor resistance; n is the total turns of coil winding; $\rho_{cu}$ is copper conductor resistivity; l is the average length of copper conductor per turn; $A_{cu}$ is the cross-sectional area of the copper conductor.

#### 3.2.2. Iron Loss

Iron loss refers to the power consumption of an electromagnet iron core under the action of an alternating magnetic field. Iron loss includes hysteresis loss, eddy current loss and residual loss [29]. The formula for calculating the hysteresis loss of ferromagnetic materials is as follows [30]:(6)$$P_{h}=\sigma_{h}fB_{m}{}^{1.6}V_{Fe},$$

Here, $\sigma_{h}$ is the hysteresis constant, the magnetic loss and the hysteresis value obtained by measuring the magnetic properties of the silicon sheet are about 300; f is frequency; $B_{m}$ is maximum flux density; $V_{Fe}$ is the volume of an iron core.

The time-varying magnetic field produces induced electromotive force in the conductor and then induced current. When the induced current forms a closed circuit in the conductor, it will produce a loss in the conductor. This induced current is called eddy current. For a single silicon steel sheet, the eddy current loss in the volume is as follows:(7)$$P_{W}=\int_{0}^{\frac{a}{2}}\frac{4\pi^{2}f^{2}bhB_{m}{}^{2}}{\rho }l^{2}dl=\frac{\pi^{2}f^{2}a^{2}B_{m}{}^{2}V}{6\rho },$$

Here, V=abh is the volume of a single silicon steel sheet; a, b and h are thickness, length and height, respectively; l is the distance from the Y coordinate. It can be seen from this formula that the eddy current loss is directly proportional to frequency f, thickness a of silicon steel and the amplitude $B_{m}$ of magnetic flux density and inversely proportional to the resistivity ρ of silicon steel sheet.

## 4. Modeling and Simulation

### 4.1. Model Equivalence

Due to the complexity and diversity of the material properties of the suspension electromagnet in the MSSS, the accuracy of the finite element model is one of the factors affecting the accuracy of the temperature field analysis results. Therefore, it is necessary to simplify the model reasonably.

Hashin and Shtrickman [31] proposed a simplified homogeneous model that equivalent the coil winding to a combination of insulating layer, impregnated paint and copper conductor. In this paper, the equivalent method of copper insulation (which does not contain epoxy resin) in the winding is regarded as another equivalent method of heat conduction. The equivalent model is shown in Figure 4.

In order to reduce eddy current loss, the iron core of the suspension electromagnet is made of silicon steel sheets with a thickness of 0.35 mm stacked one by one, and colorless transparent insulating paint is coated between the sheets. When establishing the model of temperature field simulation analysis, the silicon steel sheet and insulating paint need to be equivalent to a whole, and the equivalent model is shown in Figure 5.

### 4.2. Thermal Parameter Calculation

#### 4.2.1. Calculation of Thermal Conductivity

In the MSSS, the heat conduction medium mainly includes silicon steel sheet, coil winding and insulating material. Determining the thermal conductivity of these materials is the premise of analyzing and studying the temperature field of the MSSS.

The calculation formula of equivalent thermal conductivity of iron core silicon steel sheet is as follows [32]:(8)$$\lambda =\frac{\delta_{Fe}+\delta_{0}}{\frac{\delta_{Fe}}{\lambda_{1}}+\frac{\delta_{0}}{\lambda_{0}}},$$

Here, $\delta_{Fe}$ is the core lamination thickness; $\delta_{0}$ is the thickness of insulating paint between iron core laminations; $\lambda_{1}$ is the thermal conductivity of silicon steel sheet in iron core; $\lambda_{0}$ is the thermal conductivity of laminated insulating paint for the iron core.

The iron core of the suspension electromagnet in the magnetic suspension support component analyzed in this paper is composed of non-oriented silicon steel sheets with a thickness of 0.35 mm. Its thermal conductivity is: $\lambda_{1}=5.5\;W/\left(m\cdot K\right),\;\lambda_{y}=\lambda_{z}=44.5\;W/\left(m\cdot K\right)$.

After simplified treatment, the calculation formula of the equivalent thermal conductivity of the insulating layer is as follows:(9)$$\lambda_{j}=\frac{\sum_{i=1}^{n}\delta_{i}}{\sum_{i=1}^{n}\frac{\delta_{i}}{\lambda_{i}}},$$

Here, $\lambda_{j}$ is equivalent to thermal conductivity of insulating materials; $\delta_{0}\left(i=1,2,3,\ldots ,n\right)$ is the equivalent thickness of insulating material; $\lambda_{i}$ is the thermal conductivity of insulating materials; n is number of insulating materials.

The specific heat capacity of the equivalent insulating layer can be calculated according to the following formula:(10)$$c_{j}=\frac{c_{1}\rho_{1}V_{1}+c_{2}\rho_{2}V_{2}}{\rho V},$$

Here, $c_{j}$ is equivalent specific heat capacity of the insulating layer; V is the volume of equivalent insulation; ρ is the density of equivalent insulating layer; $c_{1}$, $\rho_{1}$ and $V_{1}$ are the specific heat capacity, density and volume of copper conductor insulation, respectively; $c_{2}$, $\rho_{2}$ and $V_{2}$ are the specific heat capacity, density and volume of epoxy resin between conductors, respectively.

Hashin and Shtrickman [33,34,35] put forward the equivalent idea of the coil winding homogenization model, and the simplified calculation formula of equivalent thermal conductivity is as follows:(11)$$\lambda_{e}=\lambda_{r}\frac{(1+k)\lambda_{Cu}+(1-k)\lambda_{r}}{(1-k)\lambda_{Cu}+(1+k)\lambda_{r}},$$

Here, $\lambda_{e}$ is effective thermal conductivity; $\lambda_{r}$ is the thermal conductivity of impregnated paint; k is the duty cycle factor of coil winding; $\lambda_{Cu}$ is the thermal conductivity of the copper conductor.

According to the above analysis and calculation, the thermal parameters of magnetic suspension support components are obtained, the specific values are shown in Table 1.

#### 4.2.2. Calculation of Heat Dissipation Coefficient

The factors affecting the flow and the thermophysical properties of the fluid itself are the two main factors affecting the convective heat transfer coefficient. For the electromagnet in the MSSS, it belongs to natural convective heat transfer. The cooling medium is calm air, so the convective heat transfer coefficient is the convective heat transfer coefficient of the electromagnet in calm air; the following table shows the convective heat transfer coefficients of heating surfaces of several materials in calm air, as shown in Table 2. According to the following table, the convection heat transfer coefficient of the electromagnet core is taken as $14.2\;W/\left(m^{2}\cdot K\right)$. The convection heat transfer coefficient of coil winding is $13.3\;W/\left(m^{2}\cdot K\right)$. According to the calculation of the convective heat transfer coefficient in document [18], the convective heat transfer coefficient of the surface between coils is taken as $9.7\;W/\left(m^{2}\cdot K\right)$.

### 4.3. Simulation Analysis

The thermal analysis module of ANSYS Workbench software is used to analyze the temperature field of electromagnet, the main component of magnetic suspension support. Firstly, the thermal analysis module and Maxwell 2D module are coupled with each other, and various losses calculated by electromagnetic calculation in Maxwell 2D are extracted as thermal loads for solution. The data between the two modules can be exchanged and updated with each other so that more accurate results can be obtained; the coupling framework of magnetic thermal analysis is shown in Figure 6. Then, select the meshing element, set the number of meshes and mesh the electromagnet finite element analysis model. The results are shown in Figure 7. The accuracy of finite element analysis is affected by the number of meshes, but at a certain number of grids, the simulation results does not change with the change of the number of meshes, so it is necessary to verify the independence of meshes. When the number of meshes is less than $150\times 10^{4}$, the temperature simulation result is not accurate, while when it is greater than $150\times 10^{4}$, the temperature result is stable and will not be affected by the number of grids.

When the current of the coil winding is taken as 0.6 A, now only the copper loss heat generation rate of the electromagnet is loaded into the finite element analysis software, and the ambient temperature is set to 30 °C. The finite element model under this condition is solved to obtain the cloud diagram of the temperature field distribution of the electromagnet when only the copper loss is considered, as shown in Figure 8.

It can be seen from Figure 8 that the temperature difference between the electromagnet core and the coil winding is not very large, and the heating at the coil winding is the most serious, with a maximum temperature of 72.1 °C. Because the temperature inside the coil winding is not easy to disperse, the maximum temperature is inside the coil winding. From the cutaway temperature distribution cloud diagram, it can also be seen that the temperature inside the coil winding is higher than that on the outer surface. When considering copper loss and iron loss at the same time, analyze the temperature field of the electromagnet. The current of coil winding is still taken as 0.6 A. Load the copper loss heat generation rate and iron loss heat generation rate into the finite element model, set the ambient temperature to 30 °C, solve the finite element model and obtain the simulation results of temperature field of electromagnet under this condition, as shown in Figure 9.

According to Figure 9, it can be seen that there is a certain difference between the temperature field distribution of electromagnet when considering copper loss and iron loss at the same time and when considering copper loss only. Since the heat generation rate of iron loss is close to that of copper loss at this time, the iron loss plays an important role in the influence of the whole temperature field. The maximum temperature is 98.9 °C. In the middle pole of the electromagnet, analyze the reasons for the temperature field distribution. On the one hand, there may be iron loss in the iron core, and at the same time, the heat of some coil windings is transmitted to the iron core through heat conduction. On the other hand, due to the limitation of the position of the middle pole, its position space is narrow and its heat dissipation coefficient is small, so its heat cannot be dissipated, so the temperature gradually accumulates. Eventually, the temperature of the middle pole is slightly higher than that of the two poles.

Considering the actual situation, the temperature field of the electromagnet with suspension support is analyzed. When establishing the simulation model, the small parts such as cylindrical pin, screw and nut are omitted, and their influence on the temperature field of the electromagnet is ignored. When the current of the coil winding is taken as 0.6 A and the ambient temperature is set as 30 °C, only the copper loss heat generation rate is applied to the analysis model, and the finite element model under this condition is solved to obtain the temperature distribution cloud diagram, as shown in Figure 10.

According to the above analysis process, by changing the current and keeping the other conditions unchanged, the model is parameterized and simulated to obtain the maximum temperature value of the electromagnet with suspension support under different currents, as shown in Table 3.

It can be seen from Figure 10 that when the suspension electromagnet is fixed on the suspension support and the temperature field is analyzed, the maximum temperature is still at the coil winding, and the maximum temperature is 67.3 °C. From the temperature distribution of the suspension electromagnet on both sides, when there is an electromagnet shell outside the electromagnet, the temperature of the electromagnet will be slightly lower. The reason is that the material of the electromagnet shell is aluminum, which has good thermal conductivity. The temperature of the electromagnet can be quickly transmitted to the electromagnet shell, and the temperature reaching the electromagnet shell is transmitted to the air by convection heat dissipation. Therefore, when the electromagnet shell exists, the temperature of the electromagnet is slightly lower than that without the electromagnet shell.

It can be seen from Table 3 that with the increase in current, the maximum temperature of the suspension bearing also increases. When the current increases to 0.8 A, the maximum temperature is 91.6 °C. Therefore, when meeting the requirements of suspension force, reduce the current of coil winding as much as possible so as to reduce the temperature of the magnetic suspension bearing system.

## 5. Experiment

### 5.1. FBG Layout and Experimental Platform Design

Firstly, the quasi-distributed temperature detection system based on FBG is designed. As shown in Figure 11, the system is divided into three modules: (1) the quasi- distributed temperature sensor network composed of FBG; (2) the transmission module composed of optical fiber jumper; (3) the sensor demodulator and computer. The quasi-distributed temperature sensor network has multiple independent optical signal demodulation channels. Many FBG temperature measurement points can be distributed on each optical fiber channel (the number of FBGs is determined by the temperature measurement range and the number of sites to be measured). Each FBG can independently measure its surrounding temperature.

When the FBG temperature sensor is used to measure the temperature of the MSSS, the number of temperature measurement points is related to the accuracy of the prediction results of the thermal analysis model. If more temperature measurement points are selected, it will increase the difficulty of measurement. If fewer temperature measurement points are selected, some important information points may be omitted, which is not enough to reflect the current temperature field; it is very important to select an appropriate number of temperature measurement points and the location of measurement points in the process of experiment. The following principles shall be followed when arranging the measuring points of the FBG sensor:(1)The measuring point position of the sensor shall be close to the heat source as far as possible and only affected by the heat source.(2)During the experiment, arrange as many temperature measurement points as possible to avoid missing important information when obtaining the best measurement points and numbers.

According to the above principles, select representative location points and reasonably arrange temperature measurement points. The selection of measurement points is shown in Figure 12. The wiring mode of the FBG sensor at the edges and corners of the iron core is shown in Figure 13.

The FBG temperature sensor is arranged on the iron core and coil winding of the E-type suspension electromagnet in the magnetic suspension support device. The sensor with multiple grid areas engraved on an optical fiber and combined with the quasi-distributed temperature measurement system can realize the “one-line and multi-point” monitoring of the FBG sensor.

The suspension bearing temperature monitoring experimental platform is mainly composed of a magnetic suspension bearing device, FBG temperature sensor, FBG demodulator, infrared thermal imager, linear DC regulated power supply and computer. The temperature monitoring experimental platform of magnetic bearing is shown in Figure 14.

In the process of applying different currents to E-type suspended electromagnetic iron, the central wavelength of the FBG temperature sensor changes with the increase in temperature. The changed wavelength is collected by the FBG demodulator, and the collected data are stored and displayed by computer. Since the FBG temperature sensor was calibrated in advance, the obtained wavelength signal can be transformed into a temperature signal to obtain the temperature value at the measuring point.

### 5.2. Temperature Measurement Experiment of Magnetic Suspension Bearing under Different Current

The temperature of a magnetic suspension bearing system is mainly determined by the temperature of the E-type suspension electromagnet. In order to obtain the temperature distribution of the E-type suspension electromagnet in a suspension bearing system under different currents, the temperature of the E-type suspension electromagnet under five different currents is measured. Apply 0 A, 0.2 A, 0.4 A, 0.6 A and 0.8 A currents to the coil winding. When the temperature of the E-type suspension electromagnet basically does not change, use the FBG demodulator to collect data for the central wavelength of each sensor on the iron core and coil winding at this time. Repeat the above operation three times.

During the experiment, several measuring points are arranged on the iron core and coil winding of the suspended electromagnet in the MSSS, and their positions are slightly different from those of the theoretical design, but they are basically on the same plane as those of the theoretical design. In the actual wiring, there are 24 measuring points on the coil winding and 18 measuring points on the iron core. Next, the data collected by the FBG temperature sensor of coil winding and iron core are analyzed. The data in the analysis process are the average value of three repeated experiments.

(1) Data analysis of FBG temperature sensor in coil winding

During the experiment, three FBG temperature sensors are arranged on the coil winding part, and each has eight grid areas, namely eight FBG temperature sensors, which are respectively recorded as FBG1–FBG8 (shown in Figure 15). In Figure 16a–c are the variation curves of temperature measured by the three FBG temperature sensors on the coil winding part with the current.

As can be seen from Figure 16a–c, under the same current, the temperature of FBG at different positions on the same optical fiber, i.e., FBG1–FBG8, is almost the same; Under different currents, the temperature of FBG on the same optical fiber has the same trend with the current; for FBG that is not at the symmetrical position on the same optical fiber, as shown in FBG1 in Figure 16a and FBG1 in Figure 16b, the temperature value displayed when the current is the same is almost the same. It can be seen from the analysis that the temperature measured by FBG at different positions on the outer surface of coil winding is almost the same within the allowable error range.

(2) Data analysis of FBG temperature sensor in iron core

During the experiment, a total of 7 FBG temperature sensors were arranged in the iron core, including 17 FBG temperature sensors. For the convenience of narration, the pole posts on both sides of the E-type electromagnet are marked as pole post 1 and pole post 3, the pole post in the middle is marked as pole post 2, and the part with the FBG sensor at the bottom is marked as the bottom of the iron core (shown in Figure 17). Figure 18a–d are the variation curves of temperature measured by 7 FBG temperature sensors in the iron core with current.

It can be seen from Figure 18d that the temperature variation trend of the four FBG temperature sensors on the bottom of the iron core with current is consistent, and the temperature variation curve with current is almost coincident. Compared with the variation curve of FBG temperature with the current in Figure 18a–c. Under the same current, the temperature value measured by each FBG at the bottom of the iron core is slightly less than that measured by each FBG on the pole column of the iron core. The analysis shows that due to the existence of coil winding on the pole column of the iron core, the heat of the coil winding transfers the heat to the iron core through heat conduction. Therefore, the iron core with the FBG sensor at the bottom of the iron core is directly exposed to the air. The three pole columns of the iron core are wrapped by the coil winding. Therefore, the heat dissipation at the bottom of the iron core is better than that of the three pole columns of the iron core, and the temperature at the bottom of the iron core is slightly lower than that of the three pole columns of the iron core.

The temperature distribution law obtained from the above simulation analysis is consistent with that of the electromagnet. Figure 19 shows the comparison between the maximum temperature measured in the experiment and the maximum temperature analyzed in the simulation under different currents. Table 4 shows the relative error between the maximum temperature in the experiment and the maximum temperature in the simulation under different currents.

The overall trend of the model and experimental results is consistent, and the error between them is less than 15%. Due to the insufficient accuracy of eddy current loss and iron loss calculation, there may be errors in applying excitation in finite element simulation, resulting in the above errors. Based on the conclusion that the temperature distribution law of the suspension electromagnet obtained from the above analysis is consistent with the temperature distribution law obtained from the simulation, the accuracy of the equivalent model in the simulation analysis and the reliability of FBG temperature measurement are verified.

## 6. Conclusions

In this paper, the equivalent model of magnetic suspension bearing is established, and its temperature field is simulated and analyzed by ANSYS Workbench. A quasi-distributed temperature measurement system is designed, and the FBG temperature sensor is used to measure the temperature field of the magnetic suspension bearing, making full use of the advantages of “one-line and multi-point” online monitoring of the FBG sensor. By comparing the experimental results with the simulation results, the experimental results are essentially consistent with the simulation results when the error is no more than 15%, which verifies the correctness of the equivalent simulation model and the reliability of FBG sensor temperature measurement.

## Figures and Tables

**Figure 1 sensors-22-04350-f001:**
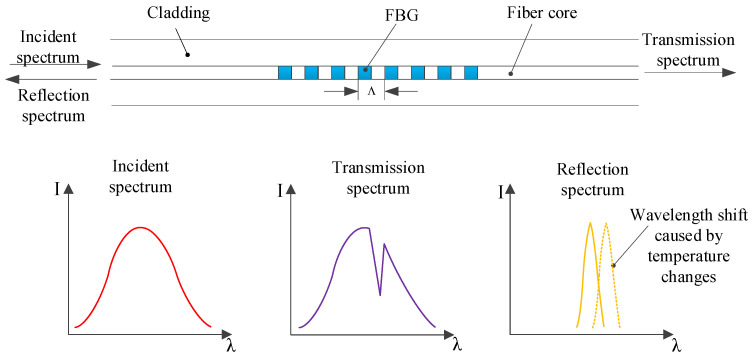
The structure and principle of FBG.

**Figure 2 sensors-22-04350-f002:**
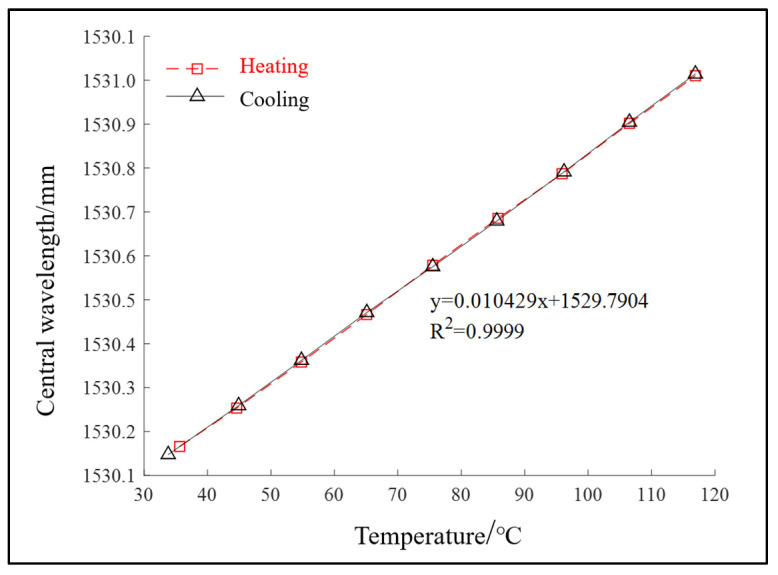
Wavelength–temperature curve.

**Figure 3 sensors-22-04350-f003:**
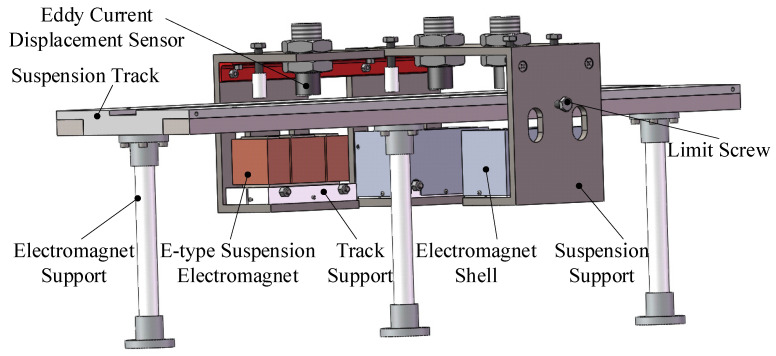
Structure diagram of magnetic bearing experimental device.

**Figure 4 sensors-22-04350-f004:**
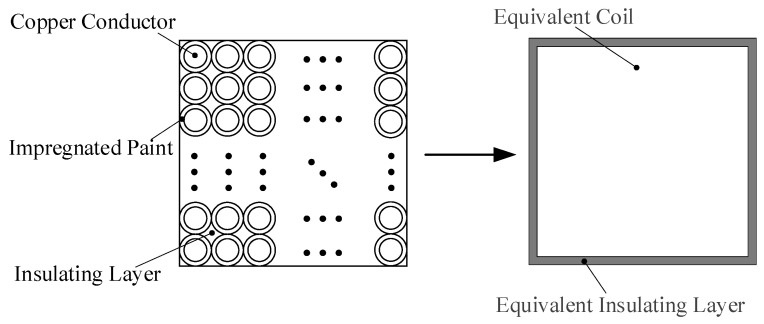
Coil winding structure and simplified model.

**Figure 5 sensors-22-04350-f005:**
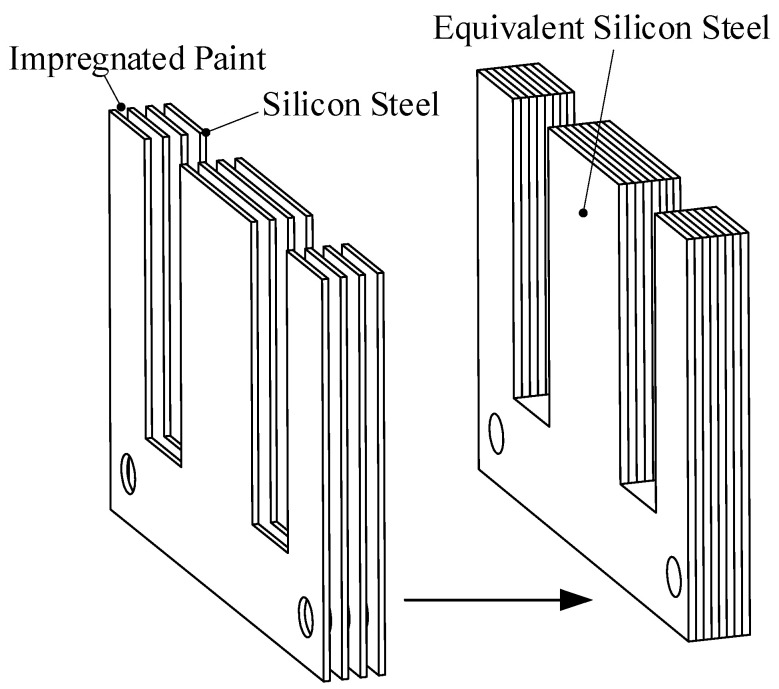
Suspended iron core structure and simplified model.

**Figure 6 sensors-22-04350-f006:**
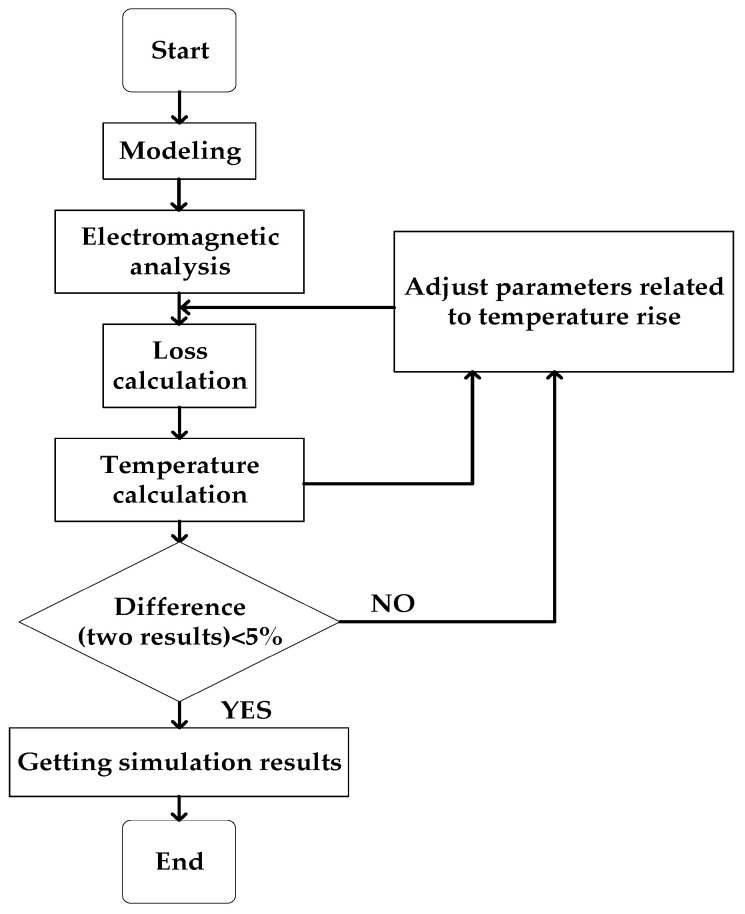
Magnetic thermal coupling analysis flow chart.

**Figure 7 sensors-22-04350-f007:**
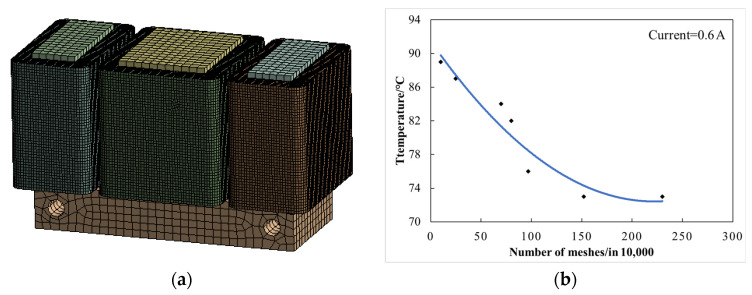
Meshing and verification diagram. (**a**) Electromagnet meshing. (**b**) Meshing independence verification.

**Figure 8 sensors-22-04350-f008:**
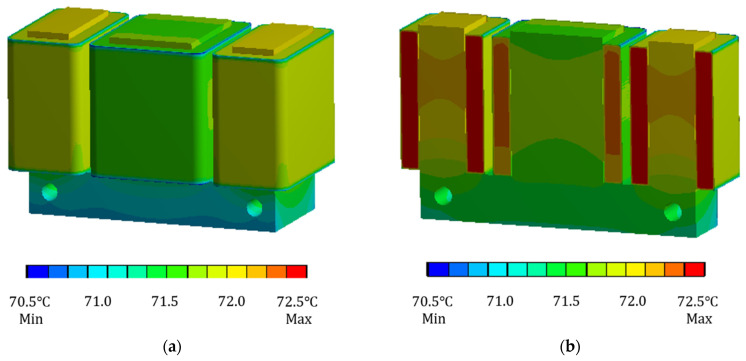
Cloud diagram of steady-state temperature field distribution of electromagnet considering only copper loss. (**a**) Overall diagram. (**b**) Cutaway diagram.

**Figure 9 sensors-22-04350-f009:**
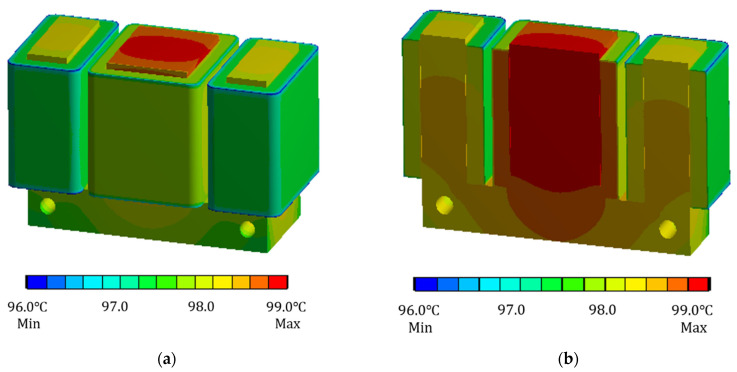
Cloud diagram of steady-state temperature field distribution of electromagnet considering copper loss and iron loss at the same time. (**a**) Overall diagram. (**b**) Cutaway diagram.

**Figure 10 sensors-22-04350-f010:**
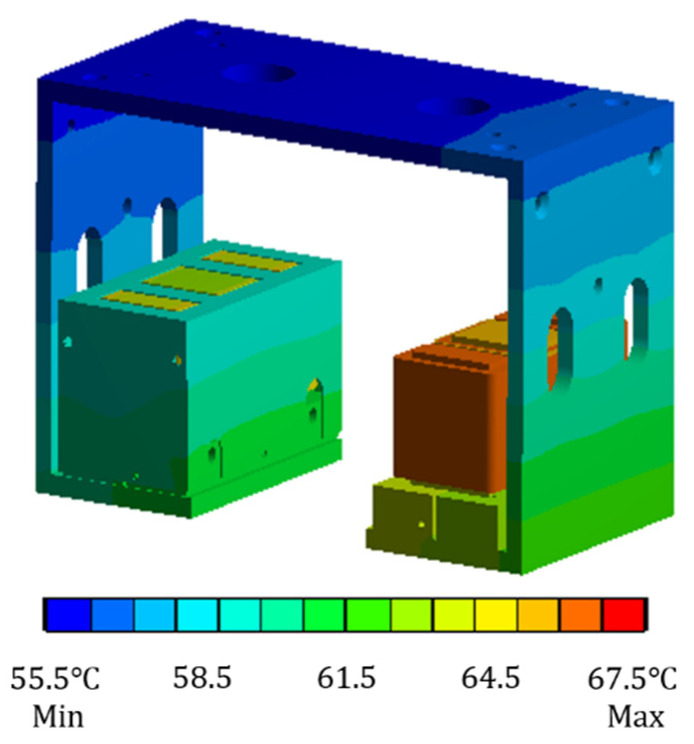
Cloud diagram of steady-state temperature field distribution of electromagnet with suspension support when only copper loss is considered.

**Figure 11 sensors-22-04350-f011:**
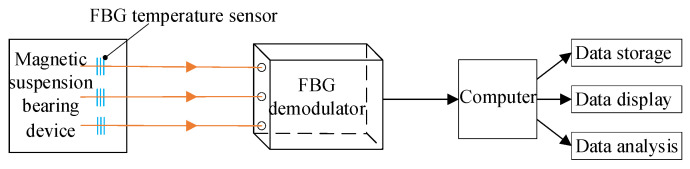
System schematic diagram.

**Figure 12 sensors-22-04350-f012:**
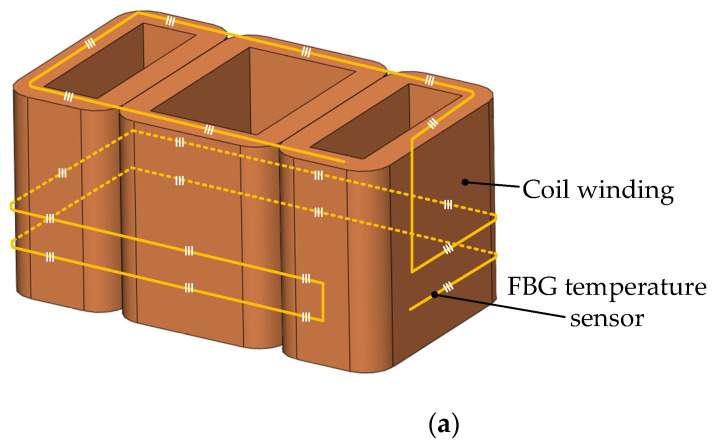
Arrangement of electromagnet temperature measuring points. (**a**) Arrangement of FBG measuring points on coil winding; (**b**) arrangement of FBG measuring points on electromagnet core.

**Figure 13 sensors-22-04350-f013:**
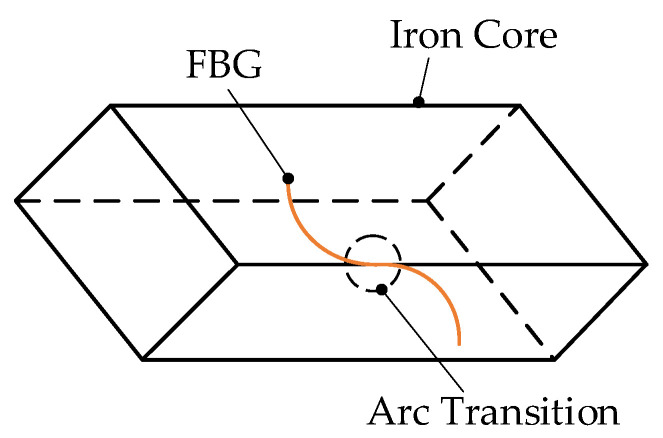
Wiring mode of FBG sensor at edges and corners.

**Figure 14 sensors-22-04350-f014:**
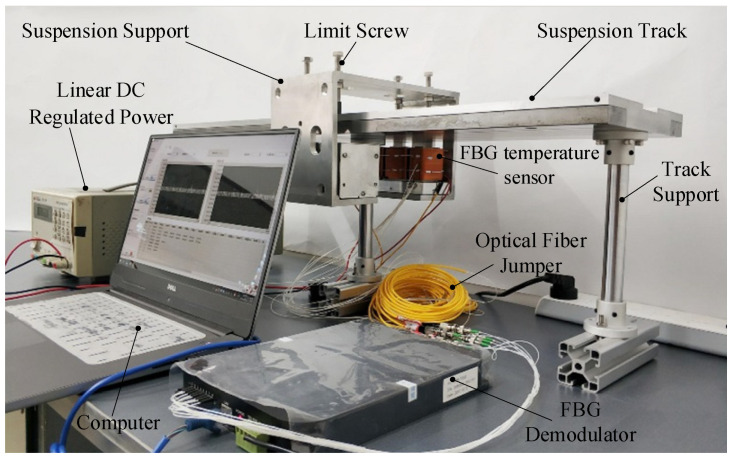
Temperature monitoring experimental platform.

**Figure 15 sensors-22-04350-f015:**
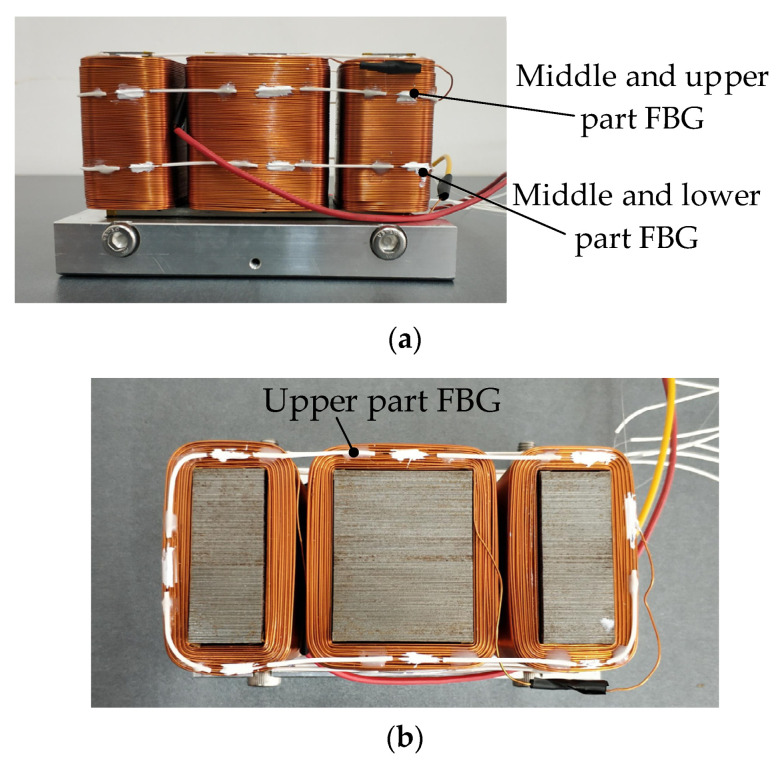
Arrangement of FBG sensor in coil winding. (**a**) Main view; (**b**) vertical view.

**Figure 16 sensors-22-04350-f016:**
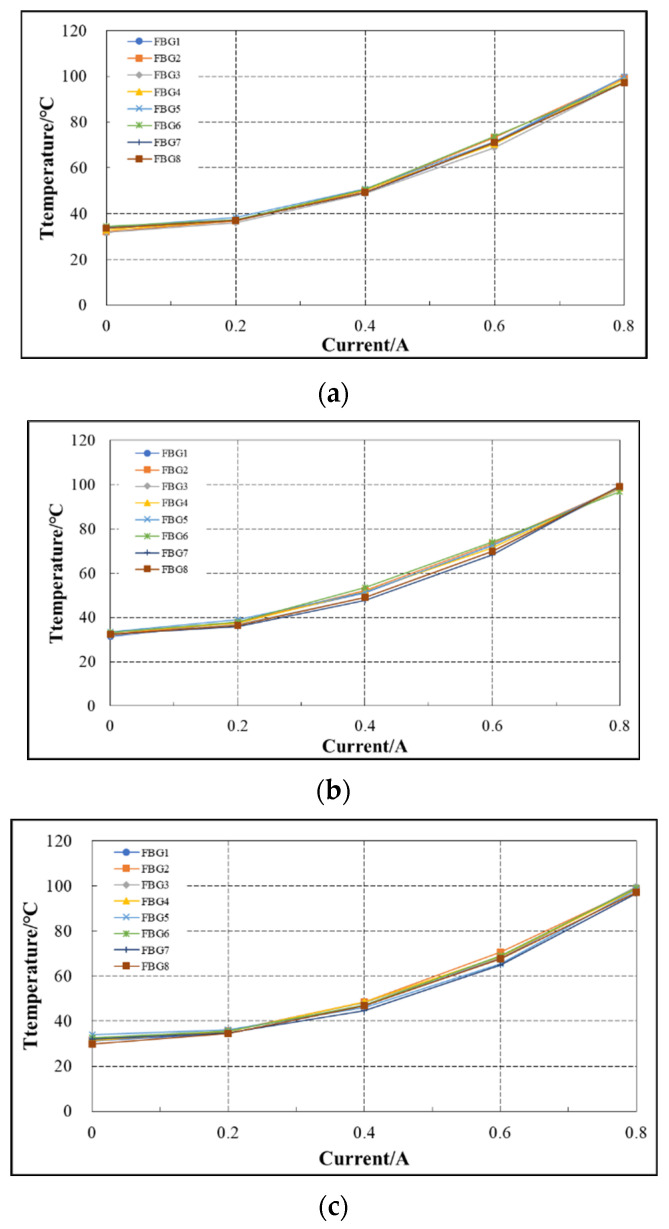
Variation curve of FBG temperature with current. (**a**) Middle and upper part of outer surface of coil. (**b**) Middle and lower part of outer surface of coil. (**c**) Upper part of outer surface of coil.

**Figure 17 sensors-22-04350-f017:**
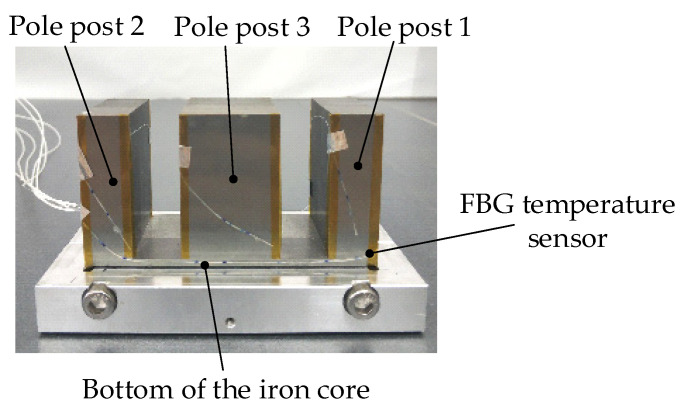
Arrangement of FBG sensor in iron core.

**Figure 18 sensors-22-04350-f018:**
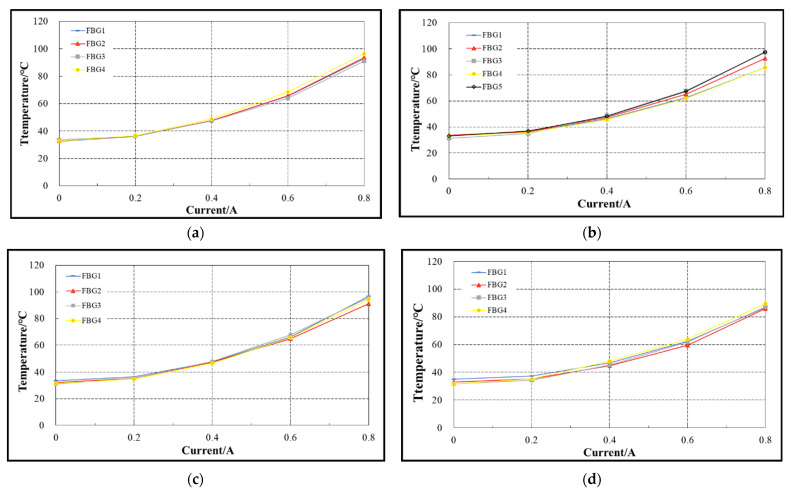
Variation curve of FBG temperature with current. (**a**) Pole post 1. (**b**) Pole post 2. (**c**) Pole post 3. (**d**) Bottom of the iron core.

**Figure 19 sensors-22-04350-f019:**
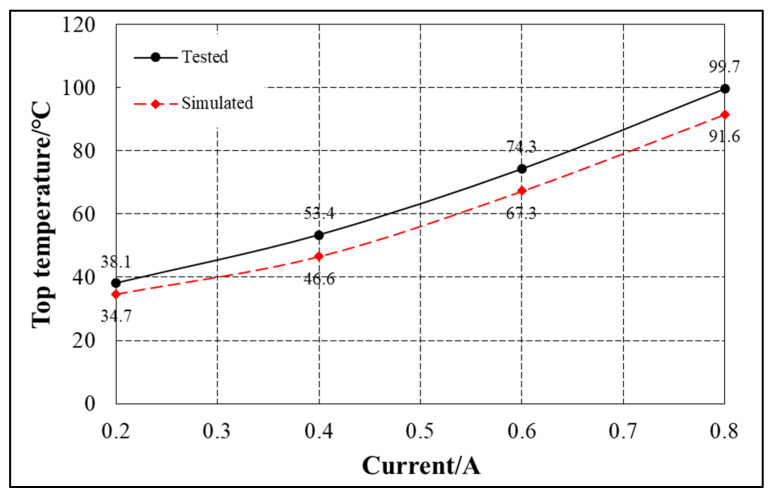
Comparison between experimental value and simulation value.

**Table 1 sensors-22-04350-t001:** Thermal parameters of magnetic bearing components.

Components	Material	$\mathrm{Thermal}\;\mathrm{Conductivity}\;\;\;(W/\left(m\cdot K\right))$	$\mathrm{Density}\;\left(kg/m^{3}\right)$	$\mathrm{Specific}\;\mathrm{Heat}\;\mathrm{Capacity}\;\;\;\;\left(j/kg\cdot k\right)$
Suspension iron	Silicon steel	$\lambda_{x}=5.5$ $\lambda_{y}=\lambda_{z}=44.5$	7650	448
Stator iron	DT4	46.5	7870	460
Coil winding	Copper	385	8920	390
Insulating layer	Impregnated paint	0.54	1672	937
Suspension support	Aluminum	237	2700	880
Electromagnet support	Aluminum	237	2700	880
Air gap	Air	0.0267	1.29	1005

**Table 2 sensors-22-04350-t002:** Surface heat transfer coefficient of different surface characteristics.

Object	$h_{0}/\left(W/\left(m^{2}\cdot K\right)\right)$
Pig iron painted with putty and paint	14.2
Painted pig iron or steel	16.7
Copper parts coated with matt or gloss paint	13.3

**Table 3 sensors-22-04350-t003:** Maximum temperature of electromagnet with suspension support under different currents.

Current/A	Temperature/°C
0.2	34.7
0.4	46.6
0.6	67.3
0.8	91.6

**Table 4 sensors-22-04350-t004:** Comparison of experimental and simulated values under different currents (maximum temperature).

Current/A	Tested/°C	Simulated/°C	Error
0.2	38.1	34.7	8.9%
0.4	53.4	46.6	12.7%
0.6	74.3	67.3	9.4%
0.8	99.7	91.6	8.1%

## Data Availability

Not applicable.

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
