# Peer review of "Simulation and Experiment Analysis of Temperature Field of Magnetic Suspension Support Based on FBG"

_sensors, 2022, doi:10.3390/s22124350_

Round 1

Reviewer 1 Report

Dear Authors,

I'm grateful to you for the opportunity to study and review your research. The material seems to me quite interesting and it would be relevant to publish it in Sensors. I have a few questions and suggestions that I listed in the file sensors-1736406-peer-review-v1 (Review).

Looking forward to your soon reply and sincerely wish you good luck!
Reviewer

Author Response

Dear Editors and Reviewers:

Thank you for your letter and for the reviewer’s comments concerning our manuscript entitled “Simulation and Experiment Analysis of Temperature Field of Magnetic Suspension Support Based on FBG” (Manuscript Number: sensors- 1736406). Those comments are all very valuable for improving our paper, as well as the important guiding significance to our researches. We have studied comments carefully and have made correction which we hope meet with approval. Revised portion are marked in red in the paper(Please see the attachment.). The main corrections in the paper and the responses to the reviewer’s comments are as follows:

Reviewer #1:

Point 1: Dear Authors, FBG sensors are known not only for their ability to accurately measure temperature, but also pressure. The FBG sensor signal significantly depends on the impact of mechanical stresses, which in this case will have a parasitic component (including residual mechanical stresses from the casing). Please tell me if you took this factor into account in your work? How and where it is necessary to take into account the additional component of the sensor error.

Response 1: In this paper, FBG packaging has adopted the way of reference 25, which eliminates the influence of stress. The influence of the temperature change of the bonding material on the measurement results is considered in advance, and the sensor is calibrated at the same time. So, the influence of mechanical stress can be obtained in advance and the error of FBG sensor can be compensated. Section 2.2 is added in the paper to illustrate, in line 160 to 186.

Point 2: Dear Authors, I read your Introduction in detail, but didn't see a brief comparative analysis for analogues of FBG temperature sensors, in addition to thermocouple and thermal resistance. There are also other temperature sensors that can measure temperature, both in direct contact and at certain distances. I'll give a couple of examples where these devices are mainly created using microelectronic technology (which is close to your Reviewer field). These are some capacitive temperature sensor https://doi.org/10.1109/LED.2020.3020937, temperature sensor based on a Schottky diode https://doi.org/10.1016/j.sna.2021.112930 and temperature sensor based on a p-n junction https://doi.org/10.1063/1.4865372. Please expand your overview in the Introduction to update these FBG temperature sensor statements. Briefly describe their advantages and disadvantages in relation to your proposed development in this application.

Response 2: Thank you for your suggested references. I have added them to the introduction.

Point 3: Why did you chose this thickness?

Response 3: In order to reduce the eddy current loss of magnetic bearing, the iron core is stacked with silicon steel sheets insulated from each other. Considering the cost and loss, the commonly used 0.35mm silicon steel sheet is selected.

Point 4: "Table 2" Move to next page.

Response 4: "Table 2" has been adjusted.

Point 5: You have indicated the dependency on the partition mesh, but you have not mentioned the dimensions. Please clarify. Also, please specify under what conditions the dependence of temperature on mesh is considered and why is it so high? I understand that these conditions will probably be discussed in the future, but this may raise questions for non-specialists.

Response 5: Meshing can adopt two methods: mesh size and mesh number. This paper is meshed by the number of meshes, so there is no need to give the size. Meshing description has been added in the paper (line 305 to 310): The accuracy of finite element analysis is affected by the number of meshes, but at a certain number of grids, the simulation results will not change with the change of the number of meshes, so it is necessary to verify the independence of meshes. When the number of meshes is less than , the temperature simulation result is not accurate, while when it is greater than , the temperature result is stable and will not be affected by the number of meshes.

Point 6: Dear Authors, you have been talking a lot about the entire Magnetic Suspension Support system, but not about the FBG temperature sensors that you use. Their own design and output characteristics are not given. Please clarify.

Response 6: "FBG temperature sensor " has been marked in Figure 14. it is explained in the Section 2.2 of the paper, in line 160 to 186.

Point 7: Please show their location, how it was done, for example in Figure 3.

Response 7: Figure 15 has been added to illustrate the installation position of FBG.

Point 8:"Figure 15"Move to previous page

Response 8: "Figure 15" has been adjusted.

Point 9: It isn't entirely clear how the temperature sensors are located. How far are the sensors from the coil winding? Also specify the paramount parameter for the accuracy of the sensor data - the non-linearity of the signal.

Response 9: FBG is arranged on both the surface of the iron core and the coil winding, while the winding is wound on the iron core. The figure15 and figure17 illustrate its location of the temperature sensors. Section 2.2 is added in the paper to illustrate, in line 160 to 186.

Point 10: Discuss what might be causing this difference between model and experiment.

Response 10: The overall trend of the model and experimental results is consistent, and the error between them is less than 15%. Due to the insufficient accuracy of eddy current loss and iron loss calculation, there may be errors in applying excitation in finite element simulation, resulting in the above errors.

Reviewer 2 Report

In this paper, the authors introduce the FBG method in the temperature measurement of the magnetic suspension system and it is helpful for related research. At first, they explain the principles of the FBG and the limitation of the thermocouple sensor used in magnetic levitation temperature detection. Then, the authors establish the equivalent model of magnetic suspension support, and analysis the temperature rise of the model considering the iron loss and without considering the iron loss. From the simulation results, they conclude the largest temperature rise part. At last, a quasi-distributed temperature measurement system is designed, and the FBG temperature sensor was introduced to measure the temperature of the magnetic suspension support system by “one-line and multi-point”. The manuscript has clear logic and rich content, however, there are still some problems in the manuscript as follows:

  1. The title should be changed to “Simulation and Experiment analysis of Temperature Field of Magnetic Suspension Support Based on FBG”
  2. The abstract should be reorganized as the order of the manuscript. the simulation is introduced first and the measurement is designed last.
  3. In the introduction about the background of Magnetic suspension support technology, the authors should add corresponding references. Like line 27 about the magnetic suspension train and magnetic bearing, there should be references.
  4. The authors should try to use the abbreviation of the professional terms, like the magnetic suspension support system, why not use the MSSS? On the other hand, the authors use both FBG and fiber bragg grating in this paper, which needs to be unified in the full manuscript.
  5. Line 117, the stop sign should be changed to a full stop.
  6. There should be a space between the number and unit.
  7. There should be a full stop between FBG and However in line 150.
  8. Figure 2 is similar to figure 13. It is suggested that the authors supplement Figure 2 and delete Figure 13.
  9. It is advised that the 2.2 should be placed behind the simulation results. Because the simulation results can provide reference for the position arrangement of FBG.
  10. In 3.1, the authors need to briefly introduce the principle of the experimental device.
  11. The authors need to explain the significance of the work. They only give the research content in this manuscript, but not the important significance of measuring temperature, such as the influence of temperature on suspension force and so on.

Author Response

Dear Editors and Reviewers:

Thank you for your letter and for the reviewer’s comments concerning our manuscript entitled “Simulation and Experiment Analysis of Temperature Field of Magnetic Suspension Support Based on FBG” (Manuscript Number: sensors- 1736406). Those comments are all very valuable for improving our paper, as well as the important guiding significance to our researches. We have studied comments carefully and have made correction which we hope meet with approval. Revised portion are marked in red in the paper (Please see the attachment). The main corrections in the paper and the responses to the reviewer’s comments are as follows:

Reviewer #2:

Point 1: The title should be changed to “Simulation and Experiment analysis of Temperature Field of Magnetic Suspension Support Based on FBG”.

Response 1: The word "analysis" has been added to the title.

Point 2: The abstract should be reorganized as the order of the manuscript. the simulation is introduced first and the measurement is designed last.

Response 2: The simulation part has been put in front of the measurement part.

Point 13: In the introduction about the background of Magnetic suspension support technology, the authors should add corresponding references. Like line 27 about the magnetic suspension train and magnetic bearing, there should be references.

Response 3: References have been added in the introduction, in line 28.

Point 4: The authors should try to use the abbreviation of the professional terms, like the magnetic suspension support system, why not use the MSSS? On the other hand, the authors use both FBG and fiber Bragg grating in this paper, which needs to be unified in the full manuscript.

Response 4: MSSS has been used to represent magnetic suspension support system.

Point 5: Line 117, the stop sign should be changed to a full stop.

Response 5: Modified to full stop, in line 125.

Point 6: There should be a space between the number and unit.

Response 6: The space between numbers and units in the paper has been added.

Point 7: There should be a full stop between FBG and However in line 150.

Response 7: Modified to full stop, in line 158.

Point 8: Figure 2 is similar to figure 13. It is suggested that the authors supplement Figure 2 and delete Figure 13.

Response 8: Figure 2 has been supplemented and is represented by Figure 11. Figure 13 deleted.

Point 9: It is advised that the 2.2 should be placed behind the simulation results. Because the simulation results can provide reference for the position arrangement of FBG.

Response 9: The position arrangement of FBG has been placed after the simulation results. This is also consistent with the abstract. (Section 2.2 is placed in Section 5.1)

Point 10: In 3.1, the authors need to briefly introduce the principle of the experimental device.

Response 10: The principle of the experiment has been briefly described, in line 189 to 194.

Point 11: The authors need to explain the significance of the work. They only give the research content in this manuscript, but not the important significance of measuring temperature, such as the influence of temperature on suspension force and so on.

Response 11: Two points about the importance of measuring temperature to the system have been explained in the introduction (line 31 to 36). (1) Affect the reliability of the system. When the system temperature exceeds its allowable temperature, it will lead to the aging of insulating materials and reduce its service life; (2) When the system temperature is too high, the temperature drift of the sensor affects the accuracy of displacement detection, which further affects the dynamic performance of the whole system.

Reviewer 3 Report

Dear Authors,

In this paper, the distributed temperature in the magnetic suspension with some fiber Bragg grading experimentally after its simulation. The measured values with 15% measurement error was the same as theoretical values. Would you check the embedded comments in the attach.

Author Response

Dear Editors and Reviewers:

Thank you for your letter and for the reviewer’s comments concerning our manuscript entitled “Simulation and Experiment Analysis of Temperature Field of Magnetic Suspension Support Based on FBG” (Manuscript Number: sensors- 1736406). Those comments are all very valuable for improving our paper, as well as the important guiding significance to our researches. We have studied comments carefully and have made correction which we hope meet with approval. Revised portion are marked in red in the paper (Please see the attachment). The main corrections in the paper and the responses to the reviewer’s comments are as follows:

Reviewer #3:

Point 1: Would you please add some reference to support this paragraph.

Response 1: References have been added in the introduction, in line 28.

Point 2: The FBG depends on the not only the temperature but also the strain. How do you distinguish both dependences?

Response 2: In this paper, FBG packaging has adopted the way of reference 25, which eliminates the influence of stress. The influence of the temperature change of the bonding material on the measurement results is considered in advance, and the sensor is calibrated at the same time. So, the influence of mechanical stress can be obtained in advance and the error of FBG sensor can be compensated. Section 2.2 is added in the paper to illustrate, in line 160 to 186.

Point 3: Would you please add a size of the silicon steel?

Response 3: The thickness of silicon steel sheet is 0.35mm, in line 241.

Point 4: How did you set the FBG to the sample?

Response 4: "FBG temperature sensor " has been marked in Figure 13. The figure15 and figure17 illustrate its location of the FBG.

Reviewer 4 Report

In this paper, the author presented a quasi-distributed temperature measurement system, which composed of FBG temperature sensors to measure the temperature of the magnetic suspension support. The experimental findings and deduced conclusions are novel and interesting. In my opinion, this manuscript is suitable for publication after minor revision:

1. The author said that FBG temperature sensor has not been used to monitor its temperature in the research on the temperature field of magnetic suspension support. Is there any other method to measure the temperature under the field of magnetic suspension support? And why no one uses FBG compared with such methods? Please give a brief explanation.

2. The abbreviations that first appear in the paper should have full names, such as FBG, NNSS and NSNS on page 2.

3. The format of the references needs to be unified.

Author Response

Dear Editors and Reviewers:

Thank you for your letter and for the reviewer’s comments concerning our manuscript entitled “Simulation and Experiment Analysis of Temperature Field of Magnetic Suspension Support Based on FBG” (Manuscript Number: sensors- 1736406). Those comments are all very valuable for improving our paper, as well as the important guiding significance to our researches. We have studied comments carefully and have made correction which we hope meet with approval. Revised portion are marked in red in the paper (Please see the attachment). The main corrections in the paper and the responses to the reviewer’s comments are as follows:

Reviewer #4:

Point 1: The author said that FBG temperature sensor has not been used to monitor its temperature in the research on the temperature field of magnetic suspension support. Is there any other method to measure the temperature under the field of magnetic suspension support? And why no one uses FBG compared with such methods? Please give a brief explanation.

Response 1: At present, thermocouple and thermal resistance can be used for contact temperature measurement, but the monitoring system has too many wires, which is complicated and vulnerable to electromagnetic interference. When using infrared thermal imager for non-contact measurement, only its surface temperature can be monitored. In contrast, FBG solves the shortcomings of the above sensors and is just suitable for the occasion of magnetic suspension support system with compact structure and high operation accuracy. There are also corresponding discussions in the introduction of the paper. In addition, our research group has research in magnetic levitation technology, fiber Bragg grating technology and other fields, and the interdisciplinary integration is closer.

Point 2: The abbreviations that first appear in the paper should have full names, such as FBG, NNSS and NSNS on page 2.

Response 2: The full name of FBG has been given in line 50. N and S are the directions of the magnetic pole, not abbreviations. 

Point 3: The format of the references needs to be unified.

Response 3: The format of references has been unified.
